# Shifting parental beliefs about child development to foster parental investments and improve school readiness outcomes

John A. List [1,2], Julie Pernaudet [1] & Dana L. Suskind[3]✉

Socioeconomic gaps in child development open up early, with associated disparities in parental investments in children. Understanding the drivers of these disparities is key to designing effective policies. We first show that parental beliefs about the impact of early parental investments differ across socioeconomic status (SES), with parents of higher SES being more likely to believe that parental investments impact child development. We then use two randomized controlled trials to explore the mutability of such beliefs and their link to parental investments and child development, our three primary outcomes. In the first trial (NCT02812017 on clinicaltrials.gov), parents in the treatment group were asked to watch a short educational video during four well-child visits with their pediatrician while in the second trial (NCT03076268), parents in the treatment group received twelve home visits with feedback based on their daily interactions with their child. In both cases, we find that parental beliefs about child development are malleable. The first program changes parental beliefs but fails to lastingly increase parental investments and child outcomes. By contrast, in the more intensive program, all pre-specified endpoints are improved: the augmented beliefs are associated with enriched parent-child interactions and higher vocabulary, math, and social-emotional skills for the children.

[1] Department of Economics, University of Chicago, 5757 S University Ave, Chicago, IL 60637, USA. [2] NBER, 1050 Massachusetts Ave., Cambridge, MA 02138, USA. [3] Department of Surgery, University of Chicago Medicine, 5841 S. Maryland Avenue, Chicago, IL 60637, USA. ✉email: dsuskind@surgery.bsd.uchicago.edu

Economic growth over the past 1000 years can be viewed as sporadic, but a persistent trend is that over the past few centuries, countries of the Organization for Economic Co-operation and Development (OECD) have fared better than their peers[1]. For centuries, scholars have explored theories to explain the causes of economic growth, with investment in human capital[2] representing a key tenet more recently. Interestingly, the last several decades have also witnessed a consensus of thought on human capital investment as a key driver of income growth and inequality at the individual level[3]. Indeed, within certain circles, scholars have argued that the most important investment society can make in its citizenry is to increase their investments in early childhood education[4].

What remains unsettled, however, is why such investments remain low among certain populations and what should be done to promote them. While the literature reveals that parental investments in children are one of the critical inputs in the production of child skills during the first stages of development[5,6], evidence also shows that such investments differ across socioeconomic status (SES hereafter)[7–9]. Even though these differences have been consistently observed across space and over time, serving to exacerbate the rising educational and income inequalities that are commonly observed in modern economies, we know little about the policies needed to address their underpinnings.

This paper takes a step back to examine sources of the disparate parental investments and child outcomes across SES to reveal one potential mechanisms for closing these gaps. We begin by presenting an economic model that invokes parents' beliefs about how parental investments affect child skill formation as a key driver of investments[10]. We then add empirical content to the model by focusing on the first few years of life, when parental investments in children have been found to play a crucial role[11–13]. To do so, we design two field experiments to explore if such parental beliefs are malleable, and if so, whether changing them can be a pathway to improving parental investments in young children. In the first field experiment, over a 6-month period starting 3 days after birth, we use educational videos informing parents about skill formation and best practices to foster child development. In the second field experiment, we test a more intensive home visiting program using assessment-based coaching and feedback for 6 months, starting when the child is 24–30 months old.

We operationalize our first field experiment by leveraging the health care system. More specifically, we built partnerships with 10 pediatric clinics predominantly serving low-SES families in the Chicagoland area and leveraged the early well-child visits. The intervention is easily replicable and relatively low-cost. In the second field experiment, we provide a home visitation intervention to low-SES families recruited in medical clinics, grocery stores, daycare facilities, community resource fairs, and public transportation in the Chicagoland area. In both cases, we measure the evolution of parents' beliefs about the impact of early child investments, parental investments and child outcomes at several time points before and after the interventions.

Our analyses point to several unique insights. First, we show that there is a clear SES-gradient in parents' beliefs about the impact of parental investments on child development. A second result provides evidence that these disparities matter, as parents' beliefs predict later cognitive, language, and social-emotional outcomes of their child. For instance, we find that parental beliefs about the impact of early child investments alone explain up to 18% of the observed variation in child language skills. A third insight is that those parental beliefs are malleable. Both field experiments induce parents to revise their beliefs about the impact of early child investments. Furthermore, exploiting the random information shocks generated by the experiments, we show that belief revision led parents to increase their investments in their child. For instance, we find that the quality of parent-child interaction is improved after the more intensive intervention (and to a smaller extent, after the less intensive intervention), and we provide evidence of a causal relationship with changes in beliefs about child development. Finally, we find positive impacts on children's interactions with their parents in both experiments, as well as important improvements in children's vocabulary, math, and social–emotional skills with the home-visiting program months after the end of the intervention. These insights represent a key part of our contribution as they show that changing parental beliefs about the impact of early child investments could potentially be an important pathway to improving parental investments in children and, ultimately, child outcomes.

Our work speaks to several branches of the literature. First, it contributes to the literature on parental beliefs by exploring the mutability of such beliefs. Research in Developmental Psychology has found that parental beliefs about child development can predict parenting practices, home environment and child outcomes[14,15], and also explain socioeconomic disparities in parental language inputs[16,17]. Recently, the economics literature introduced parental beliefs about skill formation as a key factor in models of human capital investments[10]. This literature shows that such beliefs differ across SES, and that they predict investments in children[18–23]. Not only do we replicate those different findings, but we additionally use two field experiments to demonstrate that parents' beliefs about the impact of parental inputs are malleable and link changes in those beliefs to changes in investments.

A further distinctive aspect of our study is the exploration of the full chain of impacts, from parental beliefs about child development to child outcomes, of two different types of parent-directed interventions. This approach allows us to focus on two types of interventions and to explore if each can change beliefs, and how those belief changes map into parental behaviors. In this way, our data suggest that smaller changes in parental beliefs about child development are not necessarily enough to induce lasting changes in parental investments and child outcomes.

By focusing on parental inputs, we also contribute to the literature on early language interventions in Developmental Psychology[24–30]. These papers show that providing parents with feedback, or coaching, regarding their linguistic inputs can enhance parent–child interactions. Results from our second field experiment confirm those findings in another population, English Language Learners in Spanish-speaking families, and we go one step further by assessing the impact of the intervention on a large spectrum of child outcomes.

The remainder of our paper is organized as follows. The "Results" section explores the roots of early socioeconomic inequalities via two field experiments, summarizing our design and our experimental results. The "Discussion" section provides concluding remarks. In the Supplementary Information, we present our economic framework that provides a critical link between parental beliefs about the impact of early investments, parental investments, and child outcomes, details on the experimental and econometric methodologies, as well as some additional results.

## Results

**Theoretical framework**. As the extant literature highlights, SES disparities permeate child outcomes and parental investments[9,31]. Understanding their origins and what mechanisms help to attenuate such disparities is critical. While scientists have made important strides to explore the prevalence and importance of those gaps, understanding how they arise and what environments

attenuate or exacerbate their prevalence is an underexplored area of research. We begin with an economic model summarized in the Supplementary Information (section 1). The role of the model is to highlight the importance of parental beliefs about the impact of parental inputs in the investment process.

The model starts by using the standard approach from the literature that shows the law of motion for skill formation depends on the stock of skills and parental investments in children. We depart from the standard approach by assuming that parental investment is not a black box, rather, it is predictable within a standard economic framework. In this spirit, instead of taking parental investments in children as given, we go one step deeper in the understanding of the drivers of early human capital formation and model parents' investments as a function of their contemporaneous and past beliefs about how parental investments affect child development. Specifically, we focus on parents' beliefs on the way different parenting practices affect the social-emotional and cognitive development of children between 0 and 5 years old. The theory, in this case, provides a direct link between those parental beliefs, parental investments, and child outcomes.

To add empirical content to the model, therefore, we must first explore whether differences in beliefs about the impact of early investments follow the same socioeconomic gradient as parental investments and child outcomes. Then, following our economic model, the experimental component must address whether parental beliefs are malleable and, if so, how changes in beliefs map into parental investment and child outcomes. We turn to those tasks now, first describing the two field experiments, then summarizing various signatures of parental beliefs about child development.

**Newborn field experiment**. In the first field experiment, which we refer to as the "Newborn program", we partnered with ten pediatric clinics serving medically underserved, underinsured, or uninsured populations in the Chicagoland area (see clinical-trials.gov, reference NCT02812017). Upon their arrival at the clinic, we worked with every parent during their first well-child visit (3–5 days after birth) on the days and times our research team was present at the clinics. In total, 475 parent–child dyads meeting our socioeconomic and health eligibility criteria were recruited and randomized into the treatment group (237 parents) or control group (238 parents). We obtained consent from the parents only (not the children). More detailed explanations about the recruitment and characteristics of the sample can be found in the "Methods" section.

Treatment group parents were asked to watch a series of four videos of ~10 min when they arrived at the clinic for the well-child visits at 1, 2, 4, and 6 months, which correspond to the immunization visits. In the control group, half of the parents were randomly allocated to a placebo intervention that consisted of watching a series of four videos about safety tips for babies at the same well-child visits (see Supplementary Information, Section 7.2 for more details). The other half did not watch any video. The two subgroups are pooled together for the main analysis and additional analyses separating the two subgroups are presented in the Supplementary Information, Section 7.2. Parents would typically watch the video in the waiting room before their appointment with the pediatrician on a tablet provided by a research assistant.

The treatment videos had two main parts: the first provides information on the role of parents in the very early stages of child development and on brain malleability/babies' capabilities, and the second provides practical tips that parents can apply in their daily routines with their baby. The emphasis of the tips was on nurturing language-rich serve-and-return interactions, which we coined as the "3T's" formula: Tuning in, Talking more, and Taking turns. Such responsive parenting has been shown to be associated with enhanced child vocabulary and overall child development[7,16,32–34]. In practice, two-thirds of the parents assigned to the treatment group watched the four videos, one-fourth watched only three videos, and around 8% watched two videos or less. Our intent-to-treat estimates will include all parents, irrespective of the number of videos they watched.

**Home-visiting field experiment**. The second field experiment, which we denote as the Home Visiting (HV) program, is a more intensive intervention, but it shares the Newborn program's focus on responsive parenting. Like the Newborn program, the HV program emphasized promoting nurturing language-rich serve-and-return interactions and utilized the 3Ts framework. For this more intensive program, we recruited parents of 24–30-month-old children in medical clinics, grocery stores, daycare facilities, community resource fairs, and public transportation in the Chicagoland area (see clinicaltrials.gov, reference NCT03076268). Among families who consented to participate in the experiment, 91 parent–child dyads met our socioeconomic and health eligibility criteria and were randomized into treatment (46) or control (45) groups (see the "Methods" section for a more detailed presentation of our sample). Note that we obtained consent from the parents only (not the children).

Parents assigned to the treatment group received two home visits a month for 6 months. Each of the 12 visits was approximately an hour long, and followed a specific curriculum designed to foster the cognitive and social-emotional development of the child by improving parents' beliefs and investments. During each visit, the home visitor would first show parents a video that covered a specific development topic (e.g., linguistic interactions, encouragement, incorporation of math into everyday routines) and would then do an activity with the caregiver to demonstrate how to put the concepts covered in the video into practice using the 3Ts as a framework. For example, they might practice "tuning in, taking turns, and talking more" about cooking a meal, demonstrating how that daily routine presents a perfect opportunity to engage with a child and introduce descriptive language and math terms. The second part of the visit consisted of providing feedback and setting goals for the next visit in terms of linguistic interactions between the caregiver and the child. Feedback and goals were based on recordings of the child's language exposure and production on a typical day via the LENA technology (see Supplementary Information, section 4). More than two-thirds of the treatment group parents went through the full series of 12 visits, and 24% had <6 visits (including no visit at all for 9% of the treatment group). Here again, we will present the intent-to-treat estimates.

To neutralize possible attentional effects (e.g., experimenter demand, Hawthorne effects, accountability, expression of concern from an authority), control group parents received a nutrition intervention. It consisted of sending them packets with information about the importance of healthy nutrition for child development, strategies for healthy eating, and meal preparation. Those packets were reviewed with the parents during short home visits taking place every 6 weeks over the course of the 6-month intervention phase of the study.

In both field experiments, we collected measures of parents' beliefs about the impact of parental investments on child development, measures of the quality of parent–child interactions, and measures of children's skills at regular time points pre- and post-intervention (see the timeline of the data collection for each experiment in the "Methods" section, and the description of

 

the measurement tools in the Supplementary Information, section 4).

**Disparities in parental beliefs about child development.** Our empirical analysis begins with an exploration of parental beliefs about child development at birth. Since our two field experiments focus on low-SES families, we augment our data with data gathered contemporaneously from a companion study[35] across various SES levels. Figure 1 summarizes our first result. Panel a compares the belief distribution of low-SES, mid-SES, and high-SES parents. The probability density functions are estimated using the kernel method. Low-SES is defined as household income below 200% of the Federal Poverty Line and the participating parent having at most a bachelors' degree. There are 235 parents in that category in our study. High-SES parents are defined as parents whose household income is above 400% of the Federal Poverty Line (128 observations), and mid-SES parents are neither low- nor high-SES parents (114 observations). Panel b of Fig. 1 shows the belief distributions across the educational attainment of the mother. Fifty-two mothers have less than a high-school diploma, 82 have a high-school diploma as their highest degree, 114 have some college education, and 231 have college degrees and beyond.

A first result that arises concerns the relationship between parents' socioeconomic status and their beliefs. Both figures paint a consistent and compelling picture: parents with higher levels of education or those from higher socioeconomic backgrounds were more knowledgeable about how parental investments affect child development. For example, in Panel a, the blue and green distributions, respectively, are shifted to the right of the pink distribution, suggesting that high-SES and middle-SES parents are more likely to believe that parental investments affect child development than parents of lower SES. Indeed, low-SES parents' beliefs are on average half a standard deviation lower than those of higher SES parents, and they tend to have higher variance (using a Wilcoxon test, we find that low-SES parents have a statistically different distribution of beliefs than both middle and high-SES parents at the 1% level ($p$-value < 0.001 in both cases)). Similarly, Panel b of Fig. 1 shows that those mothers with a college education believed parental investments affect their child's development more than mothers without a college education, and the differences are again significant at the 1% level with $p$-values < 0.001 (comparing parents who have a college degree and beyond with parents with some college education, parents with a high-school degree and parents without a high-school degree).

The socioeconomic disparities in parental beliefs in Fig. 1 provide insights into a potential driver of disparities in child development. They also provide a key insight that socioeconomic gaps in parental beliefs about child development open up very early in life, much like disparities in children's outcomes[36–39]. Investigating the mapping between parents' beliefs, investments, and child development is a critical step to understand how to better address the issue of inter-generational transmission of poverty, which we turn to next.

**Field experimental treatment effects.** Table 1 summarizes the impacts of our interventions measured on various dimensions at different time points. We display the effect sizes of the comparison of the treatment and control groups, the associated per comparison $p$-value adjusted for multiple hypotheses testing (see Supplementary Information, section 5 for a detailed presentation of our econometric model), and the number of observations used in the regression.

Experimental results show that, even though parental beliefs at baseline were comparable across treatment and control (detailed balancing checks are presented in Supplementary Table 2), both field experimental interventions have an immediate and lasting positive impact on parents' beliefs. For example, the top line in Table 1 shows that within 6 months, the treated group in the Newborn program has different beliefs about their investment's impact on child development compared to the control group, a difference that is significant at the 1% level. This statistically significant difference maintains in every time period for both the Newborn program and the HV program. Interestingly, the magnitude of the impact is approximately twice larger in the HV program compared to the Newborn program, and in both cases, tends to decline over time (for Newborn from $0.8\sigma$ at the first assessment to $0.6\sigma$ in the first 6 months after the end of the intervention, and $0.4\sigma$ after a year). Yet, its statistical significance remains at every time point.

A concern one may have is that these findings partly reflect an experimenter demand effect. We believe that the persistence of the impacts over time, the absence of changes in beliefs in the control group in both experiments, and the assessment protocol mitigate such concern (see Supplementary Information, section 8 for a more extensive discussion). Also note that we additionally collected other types of parental beliefs (see the Supplementary Information, section 4), and when making multiple testing corrections, we account for those measures as well. Those additional results are presented in the Supplementary Information, section 6.

To explore parental investments in children, we use detailed measures of the quality of parent–child interactions based on audio and video recordings (see Supplementary Information, section 4). As we are studying very young children (infants and toddlers), those measures allow us to capture subtle changes in parents' and children's behaviors that other coarser investment

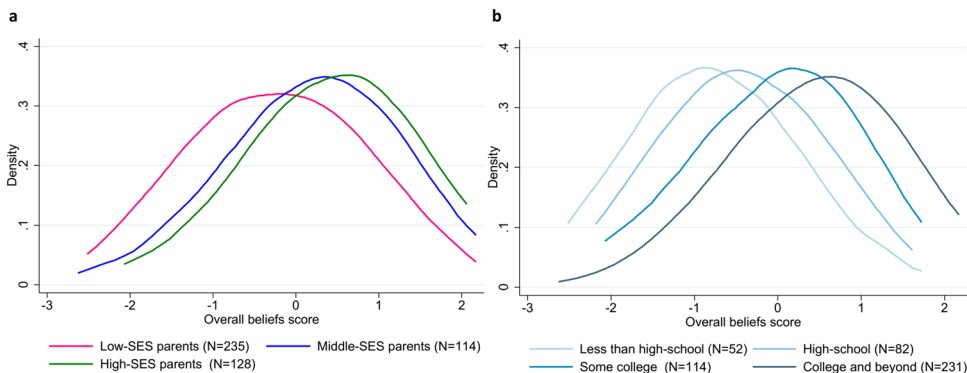

**Fig. 1 Disparities in parents' beliefs about how parental investments affect child skill formation. a** Kernel density estimation by socioeconomic status (SES). **b** Kernel density estimation by education only. Individual data points used for the estimation are provided as a Source Data file.

**Table 1 Impacts of the two experiments at different time points.**

| | Newborn program | | | | HV program | |
|---|---|---|---|---|---|---|
| | 6m | 7/9m | 12m | 18m | 30–36m | 36–42m |
| *Parents' beliefs on* | | | | | | |
| Impact of parents' inputs | 0.82*** | 0.61*** | 0.61*** | 0.38*** | 1.46*** | 1.25*** |
| | *0.00* | *0.00* | *0.00* | *0.00* | *0.00* | *0.00* |
| | 385 | 284 | 375 | 323 | 69 | 68 |
| *Parents' inputs* | | | | | | |
| Interactions w/child[a] | 0.13 | 0.27** | 0.10 | 0.10 | 0.62* | 0.44 |
| | *0.42* | *0.03* | *0.37* | *0.41* | *0.08* | *0.25* |
| | 363 | 341 | 374 | 322 | 61 | 60 |
| *Child outcomes* | | | | | | |
| Interactions w/parent[a] | −0.05 | 0.23** | -0.13 | 0.13 | 0.50* | 0.30 |
| | *0.62* | *0.05* | *0.26* | *0.33* | *0.08* | *0.33* |
| | 363 | 341 | 374 | 322 | 61 | 60 |
| Vocabulary[a] | | 0.09 | 0.11 | −0.04 | | 0.31* |
| | | *0.35* | *0.45* | *0.58* | | *0.08* |
| | | 319 | 369 | 319 | | 61 |
| Math skills | | | | | | 0.48* |
| | | | | | | *0.05* |
| | | | | | | 66 |
| Social–emotional skills | | | | | 0.44** | 0.45* |
| | | | | | *0.02* | *0.06* |
| | | | | | 69 | 68 |

Differences between treatment and control group means are shown first for the two experiments at different time points. Below in italics are the multiplicity-adjusted *p*-values based on a two-sided Student's *t* test. The adjustment procedure follows List, Shaikh and Xu (2019). Families of outcomes for the adjustment are described in the Supplementary Information, section 5. Below the *p*-values is the number of observations used in the regression. *For multiplicity-adjusted *p*-value < 0.1, **for multiplicity-adjusted *p*-value < 0.05, ***for multiplicity-adjusted *p*-value < 0.01.
[a]Signals that the measurement tools used in the two experiments are different (see Supplementary Information, section 4). In the 7/9 months column, beliefs are measured at 7 months, and interactions are measured at 9 months.

measures commonly used in surveys (e.g., number of books or toys at home, time spent reading or playing with the child) may miss. Moreover, our measures of parental inputs are based on direct observations and blind coding, which protect them against potentially non-random measurement errors that could arise in self-reported survey measures (due to experimenter demand or halo effects, typically).

Looking at the second row of Table 1, we see that there is some evidence of investment improvements. For example, for the Newborn program, the coefficient on parent/child interactions is always positive and reaches significance at the 5% level at the 9 months assessment. For the HV program, we find further evidence of improvements in the parent/child interactions, as a one-sided alternative is significant at the 5% level at our first measurement time point. Taken together, we interpret these estimates as providing suggestive evidence that our treatments, which are found to affect beliefs, also affect parental investment in children. In the Supplementary Information (section 2), we examine further the causal relationship between the changes in parents' beliefs and the changes in their investments using a two-stage least squares (2SLS) strategy. The strategy consists of leveraging the random variation in beliefs induced by the interventions to neutralize the effect of confounding factors. Our results provide evidence that the improvements we observe in parents' investments at 9 months in the Newborn experiment and 30–36 months in the home-visiting experiment are driven by changes in beliefs about the impact of parental inputs induced by the interventions.

Finally, we explore the impacts of our interventions on child outcomes. Summary results are contained in the bottom panel of Table 1. Empirical results suggest that there is some movement in child outcomes with the Newborn program, but the evidence is stronger for the HV program. For the Newborn program, the behavioral impacts are null right after the intervention, when the child is 6 months old, but become statistically significant 3 months later for interactions with parents, when the child is 9 months old

(the *p*-values are around 5% at the family level). Improvements are around +0.2σ, but they fade by the 12 and 18 months assessments. Children's vocabulary, as measured by the MacArthur–Bates Communicative Development Inventories (see the Supplementary Information, section 4), is not affected by the intervention overall. By contrast, with the HV program, children's interactions, vocabulary, math skills, and social–emotional skills are all positively impacted by treatment. Indeed, for each, we find statistically significant differences between the treatment and control families even after adjusting for multiple testing. There are several potential explanations for why the HV program moves child outcomes more than the Newborn program, but our preferred interpretation is that HV is a more intensive program compared to the Newborn program.

In the Supplementary Information, we show that the results presented in Table 1 are robust to the inclusion of control variables in the estimation (see section 7.1 of the Supplementary Information for more details).

As previously mentioned, for the Newborn program, Table 1 shows the results of the comparison between the intervention group and the two control subgroups combined (safety video and no video at all). Separating the two control subgroups confirms that the impacts on parental beliefs and inputs presented in Table 1 are driven by the content of the intervention video. The only outcome significantly affected by the safety video is the measure of children's interactions with their parents at 9 months. Both the intervention video and the placebo safety video positively improve children's interactions, leading the coefficient presented in Table 1 to be lower than when the intervention video is compared to no video at all (see the detailed results of the separate comparisons in the Supplementary Information, Section 7.2).

**Parental beliefs' predictive power**. We close our analysis with an exploration of the predictive power of parental beliefs about child

development. To do so, we augment our data with those from a companion study that follows parents and their child over a period of 2 years, starting when the child is 13–16 months old, allowing us to examine the relationship between parental beliefs and child outcomes longitudinally. The study also started in 2016 among low-SES parents from the Chicagoland area[28] (see clinicaltrials.gov, reference NCT02216032). The data contain the same measure of social-emotional skills as in the HV program, as well as two measures of language skills—the Preschool Language Scale and the Peabody Picture Vocabulary Test (see Supplementary Information, section 4). Those skills were regularly assessed over the 2 years, along with parents' beliefs about the impact of parental investments on child development.

Table 2 presents the results from simple linear regressions of child skill measures on parental beliefs. For each regression, we present the coefficient of the belief measure, its standard error in parentheses below, the $R$-squared in italics, and the number of observations. A first result is that the correlations between parental beliefs and children's skills are all positive and mostly significant at 1%, both across different ages and across different skill measures. The strongest correlations are for linguistic skills (Preschool Language Scale (PLS) and Peabody Picture Vocabulary Test (PPVT)), which is not surprising given the emphasis on the role of parent talk in our belief survey.

Empirical results shown in the first column of Table 2 indicate that a $1\sigma$ increase in parental knowledge when the child is between 13 and 16 months old is associated with a $0.8-1.2\sigma$ increase in language skills later in life. The next columns reveal that this relationship can be even stronger at later stages, when the child is 19–22 to 37–40 months old. Considering the $R$-squared value, we find it important that beliefs alone explain up to 18.7% of the variation in child language skills (for beliefs measured at 19–22 months and language skills measured a year and a half later with the PLS) and around 13–15% for most other ages as well as for our other measure of language skills that focuses on vocabulary (PPVT, last row of the table).

## Discussion

The literature on child development has taught us that early childhood investments map into long-term outcomes[40,41], calling for more research on the optimal policies required to reduce early inequities within modern societies. Yet, there is much heterogeneity in childhood investment and child outcomes observed across socioeconomic strata. We approach early childhood disparities differently in that we take a step in the opposite direction from most and ask a basic question: what are the key drivers of parental investment in children? And, can these drivers help to uncover why we observe constant differences across socioeconomic strata?

Through a simple economic framework, we are directed to focus on parental beliefs about child development. Such an approach proves rich in that it ties parental beliefs to parental investments and, ultimately, to child outcomes. To add empirical content to the model, we change beliefs via two field experiments, and show how these changes in belief can be mapped into changes in parental investments. Importantly, we find improvements in various school readiness outcomes with our more intensive program. Though our results point to one potential cause of disparate outcomes, we recognize that changing parental beliefs will not address or overcome many of the other deep, structural drivers of inequality. Nevertheless, our results have several potential policy implications. They first suggest that providing information and guidance that can change parental beliefs about the impact of parental investments in children can be one pathway to improving school readiness outcomes among low-SES families. They also suggest that simple educational policies may not be sufficient to induce robust behavioral changes and child outcome improvements. Indeed, while both of our interventions show that parental beliefs are malleable, the more intensive program has roughly twice the impact on beliefs as our less intensive intervention. It is a question for future research to understand whether the associated differences in parental investments and child outcomes are due to the level of belief change each program caused. We view our approach as a starting point, which future programs meant to reduce

**Table 2 Predictive power of parental beliefs about child development for children's skills.**

| | Beliefs on impact of parents' inputs at: | | | | | | | |
|---|---|---|---|---|---|---|---|---|
| | **13–16m** | | **19–22m** | | **25–28m** | | **37–40m** | |
| *ASQ-SE at* | | | | | | | | |
| 19–22m | 0.19* | 4.74% | 0.17* | 3.87% | | | | |
| | (0.10) | 68 | (0.10) | 77 | | | | |
| 37–40m | 0.14 | 0.99% | 0.29* | 4.23% | 0.31* | 3.86% | 0.40** | 7.76% |
| | (0.18) | 67 | (0.16) | 74 | (0.18) | 73 | (0.16) | 76 |
| *PLS at* | | | | | | | | |
| 13–16m | 0.20** | 5.14% | | | | | | |
| | (0.10) | 83 | | | | | | |
| 19–22m | 0.84*** | 13.15% | 0.99*** | 15.35% | | | | |
| | (0.27) | 68 | (0.27) | 77 | | | | |
| 25–28m | 1.05*** | 13.24% | 1.24*** | 15.69% | 1.09*** | 9.82% | | |
| | (0.33) | 68 | (0.34) | 76 | (0.39) | 75 | | |
| 31–34m | 0.94*** | 12.22% | 1.41*** | 18.19% | 1.35*** | 13.60% | | |
| | (0.31) | 69 | (0.35) | 76 | (0.40) | 75 | | |
| 37–40m | 1.24*** | 14.40% | 1.40*** | 18.65% | 1.40*** | 15.03% | 1.25*** | 14.09% |
| | (0.38) | 64 | (0.35) | 72 | (0.40) | 70 | (0.37) | 73 |
| *PPVT at* | | | | | | | | |
| 37–40m | 0.32*** | 14.02% | 0.35*** | 15.98% | 0.35*** | 12.67% | 0.31*** | 11.66% |
| | (0.10) | 63 | (0.10) | 71 | (0.11) | 69 | (0.10) | 72 |

The rows correspond to children's skills measured at five different time points. ASQ-SE stands for Ages and Stages Questionnaire: Social Emotional, PLS stands for Preschool Language Scale, and PPVT stands for Peabody Picture Vocabulary Test (see Supplementary Information, section 4.3). They are regressed on parents' beliefs measured at four different time points, in separate simple linear regressions. Both the child outcome variables and the beliefs variables are standardized. For each regression, the table gives the coefficient of the beliefs measure, its standard error in parentheses below, the $R$-squared in italics next to the coefficient, and the number of observations below the $R$-squared. *For $p$-value < 0.1, **for $p$-value < 0.05, ***for $p$-value < 0.01. $p$-values are based on two-sided Student's $t$ tests without adjustment for multiple comparisons.

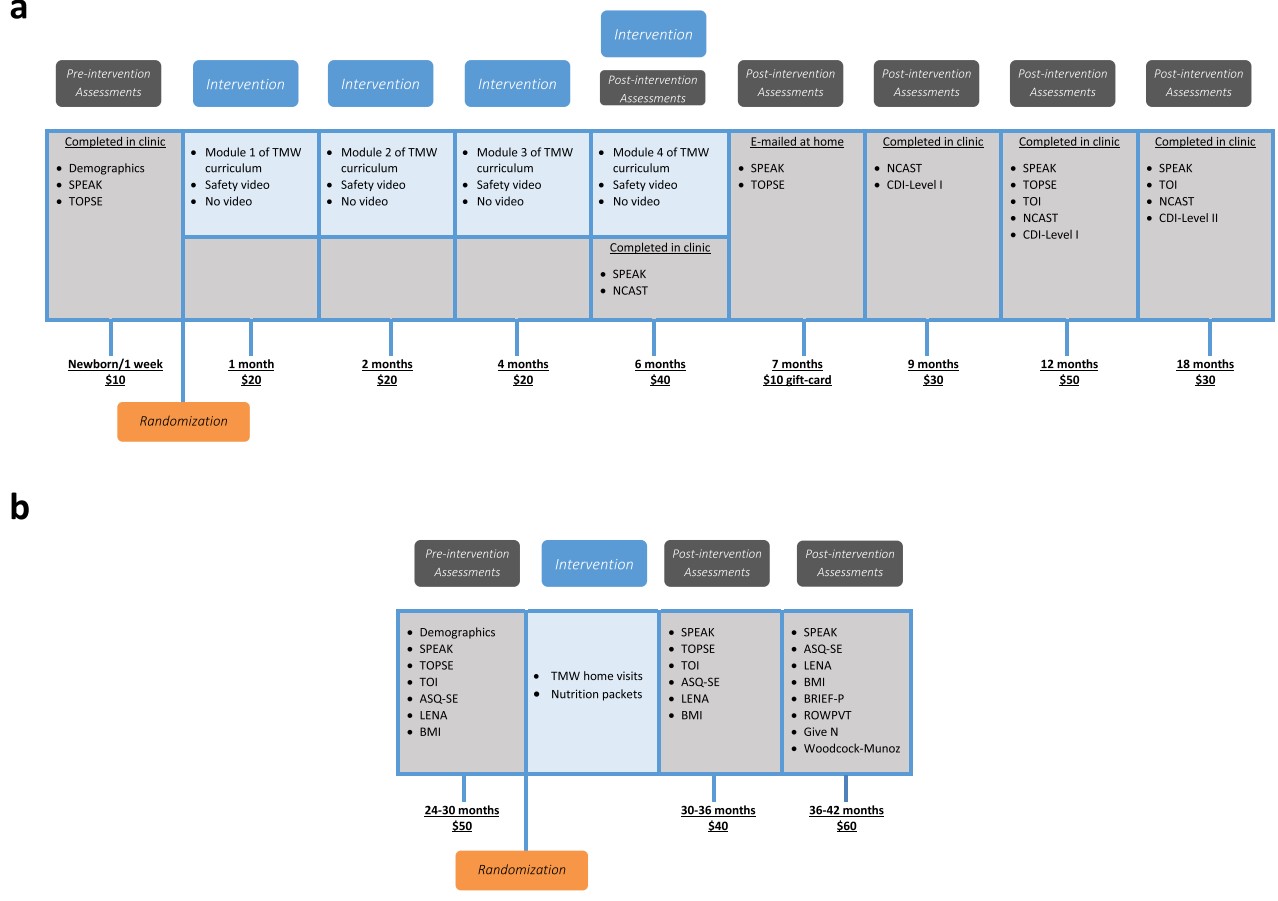

**Fig. 2 Timeline of the studies with the surveys used. a** Newborn study. **b** Home-visiting study.

informational frictions can build on to reduce socioeconomic gaps in child development.

## Methods

**Sample, randomization, and balancing checks**. In addition to the income and education criteria described in the paper, we applied age, language, and health criteria to select parents. In both experiments, eligible parents had to be female primary caregivers, at least 18 years old, had to live with their child and have legal custody, and their child had to fall into a certain age range (<30 days old for the newborn program and 24–30 months old for the home-visiting program). In the newborn program, parents had to speak English or Spanish, whereas in the home-visiting program, they had to speak Spanish as their preferred language at home. Finally, children born with significant perinatal or neonatal complications, developmental disabilities or medical problems, and before 36 weeks were excluded from the newborn study, and children with significant cognitive or physical impairments were excluded from the home-visiting study. In both experiments, parents who lived above 200% of the federal poverty line, parents whose education level was more than a bachelor's degree, foster parents, and parents who were unable to commit to the intervention requirements were also excluded. The full protocols are registered on clinicaltrials.gov, references NCT02812017 and NCT03076268.

For all the studies involved in the analysis, we received approval from the University of Chicago Biological Sciences Division Institutional Review Board. The experiments complied with all relevant ethical regulations for work with human participants and involved informed consent of the parents that was obtained through written signature after they read the form and asked any questions they may have to a trained Research Assistant.

For the newborn study, we targeted a baseline sample size of a minimum of 400 participants based on power calculations setting a target power threshold of 80%, a significance level of 5%, and assuming a 25% dropout rate. Data from a previous study by Suskind et al. [35] were used to assess the size of expected treatment effects on parental beliefs (measured by the Survey of Parents' Expectations and Knowledge, see Supplementary Information, section 4).

For the home-visiting study, we targeted a baseline sample size of a minimum of 90 participants based on power calculations also setting a target power threshold of 80%, a significance level of 5%, and assuming a 25% dropout rate. Data from our Longitudinal home-visiting study[28] were used to assess the size of expected

treatment effects on parent–child interactions (measured by the LENA recordings, see Supplementary Information, section 4).

For the newborn study, the first enrollment took place on June 20, 2016 and the last enrollment on July 31, 2017. For the home-visiting study, the first enrollment took place on April 17, 2017 and the last enrollment on July 9, 2018.

To provide a more detailed description of each sample, Supplementary Table 2 (section 3 of the Supplementary Information) shows the average characteristics of the control group measured in our baseline surveys for each study. Participants are by construction from low-SES families: almost a third of them do not have a high-school diploma, around 80% have a household income below $2655 per month and a large share of families are enrolled in social programs such as insurance or food programs in both samples. In the newborn sample, 35% of the respondents were employed at the time of the recruitment, while this proportion is 51% in the home-visiting sample. 54% and 100%, respectively, are Hispanic or Latino in the newborn and home-visiting study, and a little <30% declare themselves White in both cases.

Parents who enrolled in the experiments were randomly assigned to the treatment or control group so that the two groups are on average similar, and a simple comparison of their outcomes post-intervention allows us to recover the causal impact of the intervention. In both experiments, the randomization was done using the website Research Randomizer to generate a list of unique unsorted numbers (91 numbers in the HV program and 475 numbers in the Newborn program), each paired with a participant number. The first half of the random sequence was then assigned to Treatment, and the second half to Control. For the Newborn program, the procedure was done separately for English-speaking and Spanish-speaking participants.

We cannot directly test whether the randomization indeed created similar groups, but we can compare control and treatment parents along their baseline characteristics to check that at least their observed characteristics are on average similar. Those comparisons are shown in the second and fifth columns of Supplementary Table 2 ($T - C$) for the set of demographic variables, a synthetic dummy built from the Family Life Events survey indicating whether the parent reported at least one traumatic event that occurred in the past 6 months (divorce, death, violent crime, serious illness, depression, prison, child lived with someone with alcohol or drug issues, extreme financial difficulties), and the belief scores.

The comparisons reveal that some variables tend to be slightly unbalanced (parent's age and the likelihood of experiencing a traumatic event in the newborn study; employment status in both studies) but the difference is generally significant only at the 10% level without accounting for multiple hypotheses testing, except for

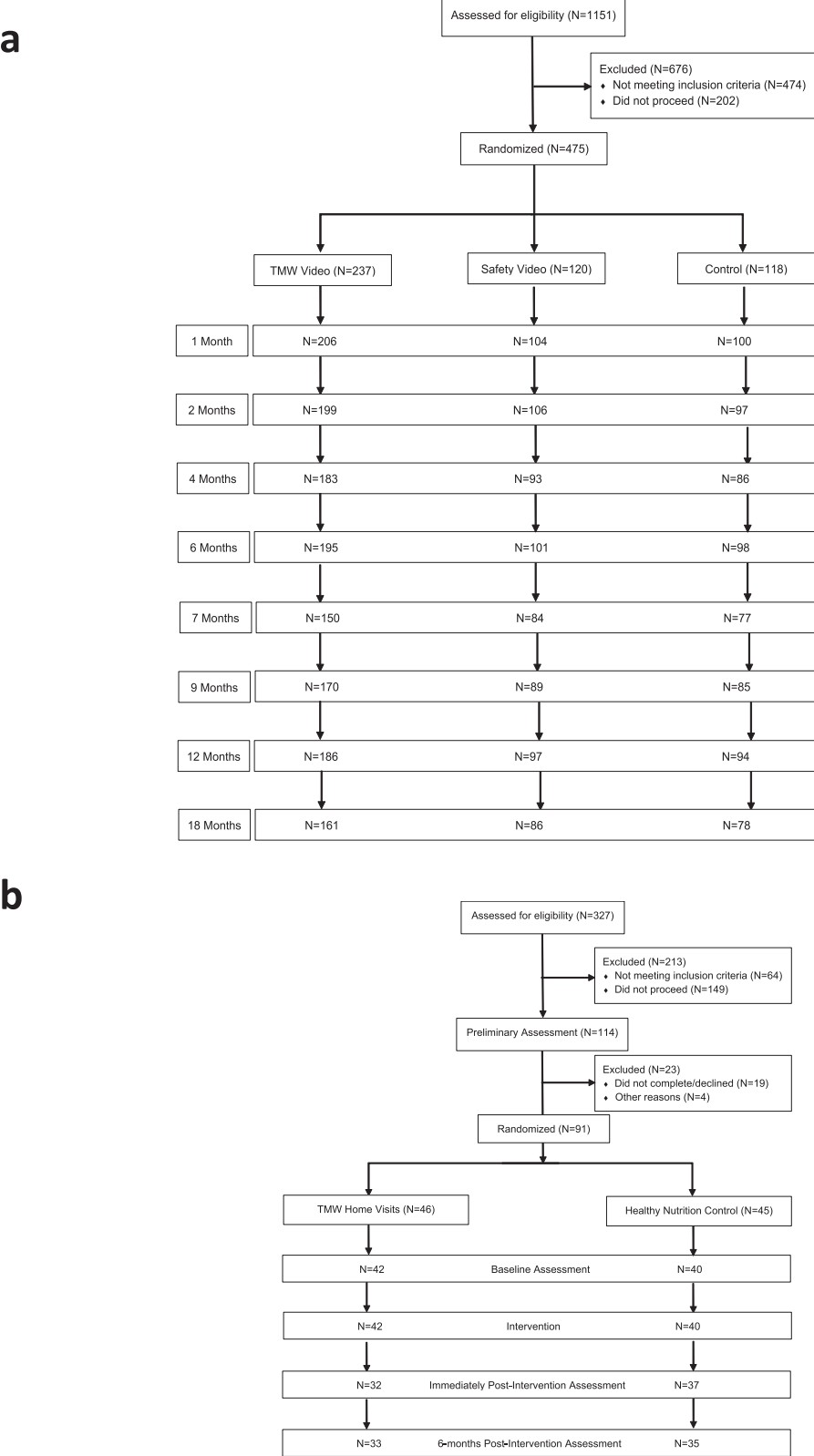

**Fig. 3 Enrollment and attrition rates at each assessment point. a** Newborn study. **b** Home-visiting study.

the employment status in the home-visiting study which is significantly different at 1%. Since we estimate impacts separately at each time point in the impact analysis and there is attrition between each time point that could generate further imbalance (if related to the treatment status), we did those comparisons separately at each time point (tables available upon request). A few variables become imbalanced at later time points in the newborn study (LINK card, number of

children). Regressions taking into account imbalances between the treatment and control groups are presented in section 7 of the Supplementary Information.

**Timeline, enrollment rates and attrition**. Figure 2 presents the timing of the different surveys parents completed and the videos they were asked to watch for the newborn study (panel a) or the home visits they received in the second study

(panel b), along with the payment they received to do it. The acronyms of the surveys are explained in section 4 of the Supplementary Information. In Panel a, the TMW curriculum modules correspond to the videos of the intervention, the safety videos are videos watched by our placebo control group (see section 7 of the Supplementary Information for explanations about the placebo control group).

Figure 3 shows the number of parents initially approached, the number of parents who were randomized, as well as the attrition over time in each group for both experiments.

The paper complies with ICMJE's reporting guidelines.

**Reporting summary**. Further information on research design is available in the Nature Research Reporting Summary linked to this article.

## Data availability

The datasets generated and/or analyzed during the current study are not publicly available due to the presence of Protected Health Information, but all de-identified datasets will be made available upon request from the corresponding author to replicate the results. Source data are provided with this paper.

## Code availability

All codes used to generate the results presented in this paper are available from the corresponding author on request. All data were collected using REDCap and analyzed using Stata 15.

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

## Acknowledgements

The different studies used in this paper have been funded by the W.K. Kellogg Foundation, the Pritzker Traubert Family Foundation, the PNC Foundation, the Heising-Simons Foundation, and the Hymen Milgrom Supporting Organization. We are grateful to Amanda Bezik, Alison Hundertmark, Kristin Leffel, Christy Leung, and Michelle Saenz for the implementation and monitoring of the projects. We also thank participants

of the ASSA 2021 meeting, the ESA 2019 conference, the AFE 2019 conference and the RCT Days 2019 conference for their useful comments.

## Author contributions

J.A.L., J.P., and D.L.S. were equally involved in the design of the experiments, data analysis, and writing of the paper.

## Competing interests

The authors declare no competing interests.
