## [Peer Review File · Nature Communications]

Shifting parental beliefs about child development to foster parental investments and improve school readiness outcomesEditorial Note: This manuscript has been previously reviewed at another journal that is not operating a transparent peer review scheme. This document only contains reviewer comments and rebuttal letters for versions considered at *Nature Communications*.

Reviewer #2 (Remarks to the Author):

The paper contributes to two strands of the literature on early childhood development. One strand is within Economics, and it investigates how heterogeneity in parental beliefs (about the impacts of investments on human capital development) predicts (or causes) heterogeneity in investments. The second literature relates to a growing field of language interventions in Developmental Psychology. The paper has two key contributions, and it should be published. However, the authors did not address two types of comments from the original set of reviewers. The first type of comment relates to the description of the paper fits into the literature in Economics and Developmental Psychology. The second type of comment relates to aspects of data analysis.

Literature:

First, it is necessary to place the paper in the two pieces of literatures to which it relates. First, the paper has a significant contribution to the literature on parental beliefs in Economics. However, the placement of the paper in this literature is very fuzzy.

1. Unlike what the authors suggest, the paper does not introduce a model of parental beliefs in human capital formation. This introduction was done elsewhere (Cunha, Elo, and Culhane, 2013).
2. The paper replicates the findings of the correlation between parental beliefs and family's SES by Boneva and Rauh (2018) and Birolli, Boneva, and Rauh (2020). The paper's citation to their work is appropriate.
3. The literature has also shown that low SES parents have beliefs about the impact of investments on human capital (i.e., beliefs about the marginal productivity of investments) that are too low in comparison to objective estimates of the marginal productivity of investments (Cunha, Elo, and Culhane, 2013; Attanasio, Cunha, and Jervis, 2019).
4. The literature has also shown that parental beliefs predict investments in children (Attanasio, Cunha, and Jervis, 2019; Cunha, Elo, and Culhane, 2020).

This paper has a significant contribution to this literature. It shows that parental beliefs are malleable. This paper is the first to find such a result. It is an important finding, and it is worthy of publication.

Second, the placement of the paper in the literature of early language interventions in Developmental Psychology is confusing. There are several recent papers in this area focusing on interventions with parents (Suskind et al., 2013; Suskind et al., 2016; McGillion et al., 2017; Leech et al., 2018; Ferjan Ramirez et al., 2020). Surprisingly, the authors cite none of these papers. The uncited papers are more similar to the home-visitation study, less similar to the newborn study because they include coaching with feedback from LENA. Also, these studies find significant impacts on parental behaviors. Some of these papers also measure parental knowledge that is very close to the SPEAK assessment tool used in the current study.

In summary, I agree with the reviewers that the authors do a poor job of summarizing how their research fits into the literature in economics and the literature in early language development. The

response from the authors is that they have little space. I think this response is not acceptable. First, Section 2 of the paper wastes space. They reproduce data from research well-known by economists and developmental psychologists working in areas of human capital formation, who are the audience for their paper. So, by saving the space in Section 2, they could use the space to have a much better description of how their work contributes to the literature. Therefore, I do not judge that the authors correctly addressed the original comments from the previous reviewers.

Data Analysis.

The results in the paper are that the newborn intervention did not produce impacts, and the writing of the paper should reflect the data presented in Table 1. There is one coefficient that is statistically significant (at 7/9 m). However, the point estimate and the standard error are very different from their counterparts at 6m, 12m, and 18m.

This non-statistical significance is scientifically significant. The newborn program is a “light-touch” intervention. Also, the newborn program does not seem to offer feedback with the LENA reports. Therefore, the statistical non-significance shows two limits of the interventions with parents to promote language development: light-touch interventions or lack of feedback may improve parental beliefs, but not affect parental investment behavior. This result is scientifically significant: the malleability of beliefs, by itself, is not enough. A more intense coaching intervention is necessary. This finding has important consequences for the design of public policy in this area. For this reason, this finding is worthy of publication, and the authors should include a thorough discussion session comparing their results with the results from the previous literature in Economics and Developmental Psychology.

The reviewers also request the authors to report the coefficients disaggregated by measure. Such a request is particularly important for the measures of parental inputs. The two measures they have (NCAST and LENA) are high-quality measures. However, the literature in early language development has reported impacts separately for the automatic counts produced by LENA because a primary target of these interventions is to increase the quantity of adult-child conversational turns.

I completely agree that the approach they take is robust to false positives. However, the disaggregation of the impacts by measurement instruments improves comparability across papers. This comparability is a legitimate scientific goal. The other papers in this literature do not necessarily assess parental inputs via NCAST. Even if they did, it is not clear that one would combine the scores of a measure of quantity (e.g., conversational turns as measured by LENA) with a measure of quality (measured by NCAST) of parent-child interactions. For this reason, I do not judge that the authors have addressed the comments on data analysis by the original set of reviewers.

Attanasio, O., Cunha, F., and Jervis, P. (2019). Subjective Parental Beliefs. Their Measurement and Role. NBER Working Paper w26516.

Birrolli, P., Boneva, T., Raja, A., Rauh, C. (2020). Parental beliefs about returns to child health investments, *Journal of Econometrics*, in press.

Boneva, T. and Rauh, C. (2018). Parental Beliefs about Returns to Educational Investments—The Later the Better? *Journal of the European Economic Association*, 16(6), 1669-1711.

Cunha, F., Elo, I., Culhane J. (2013). Eliciting Maternal Beliefs about the Technology of Skill Formation. NBER Working Paper w19144.

Cunha, F., Elo, I., Culhane J. (2020). Maternal subjective expectations about the technology of skill

formation predict investments in children one year later, *Journal of Econometrics*, in press.

Ferjan Ramírez N, Lytle SR, Kuhl PK. Parent coaching increases conversational turns and advances infant language development. *Proc Natl Acad Sci U S A*. 2020 Feb 18;117(7):3484-3491. doi: 10.1073/pnas.1921653117. Epub 2020 Feb 3. PMID: 32015127; PMCID: PMC7035517.

K. Leech, R. Wei, J. R. Harring, M. L. Rowe, A brief parent-focused intervention to improve preschoolers' conversational skills and school readiness. *Dev. Psychol.* 54, 15–28 (2018).

M. McGillion, J. M. Pine, J. S. Herbert, D. Matthews, A randomised controlled trial to test the effect of promoting caregiver contingent talk on language development in infants from diverse socioeconomic status backgrounds. *J. Child Psychol. Psychiatry* 58, 1122–1131 (2017).

D. Suskind et al., An exploratory study of 'quantitative linguistic feedback': Effect of LENA feedback on adult language production. *Comm. Disord. Q.* 34, 199–209 (2013).

D. Suskind et al., A parent-directed language intervention for children of low socioeconomic status: A randomized controlled pilot study. *J. Child Lang.* 43, 366–406 (2016).

48. M. L. Rowe, K. A. Leech, A parent intervention with a growth mindset approach improves children's early gesture and vocabulary development. *Dev. Sci.* 22, e12792 (2019).

Reviewer #3 (Remarks to the Author):

Reviewer 3

The authors have not addressed my first point about what the existing literature says about the question being considered here. I'm not suggested a substantial review of the literature on early intervention but a focused consideration of what existing studies say about the question here and what is distinctive about this study that allows it to add to existing evidence.

The authors provide some additional information on methods but they need to describe randomization - how the sequence was generated, whether there was concealment and whether there was any stratification to minimize possibility of imbalance.

1. The authors do not describe how the paper fits into the literature in Economics and Developmental Psychology

→Thanks much for this comment. We added a more detailed literature review at the end of the introduction presenting the contributions in the two disciplines. We included the references mentioned in the report and discussed points 1., 2., and 4. The qualitative nature of the questionnaire we use to elicit parental beliefs does not allow us to speak to point 3 (we cannot quantify the bias in respondents' beliefs).

We also clarified that the introduction of the model of parental beliefs was done in Cunha, Elo and Culhane (2013), following point 1.

Finally, we removed section 2 as suggested by the reviewer to save space for a more comprehensive literature review. We shifted the references to the two literatures to the beginning of the next section, as we refer later in the paper to the socioeconomic gaps in parental investments and child outcomes that used to be presented graphically in section 2.

2. The writing of the paper should reflect the data presented in Table 1

→Great point. Table 1 puts into perspective the two experiments. We adjusted the writing of the abstract and introduction to better reflect the differences between the two sets of results, highlight the non-significant results of the light-touch intervention, the fact that it does not include feedback, and draw policy implications, as suggested in your report.

3. Report the coefficients disaggregated by measure

→ In Table 1, only the two measures of interactions in the Newborn program and the vocabulary measure in the HV program are aggregated. None of the measures are aggregated in Table 2.

Table A.4 presents the disaggregated results for the Newborn program, and Table A.5. presents the disaggregated results for the HV program.

Note that we did not combine NCAST with LENA. NCAST was used in the Newborn program, and LENA was used in the HV program. In Table 1, the row "interactions w/ child" corresponds to the NCAST parental score (first four subscores) in the Newborn program, and to the CTC in the HV program. The row "interactions w/ parent" corresponds to the NCAST child score (last two subscores) in the Newborn program, and to the CVC in the HV program. The third LENA count is the AWC. We added explanations for why we excluded that count in appendix D – LENA section.

For the Newborn program, we used a measurement tool that was valid starting at age 6 months, hence the choice of NCAST.

We use the symbol ° in the table to signal that the measurement tools used in the two experiments are different and refer in the caption to appendix D, where we provide more detailed explanations about the measurement tools and construction of the scores.

We added clarifications about NCAST and LENA in appendix D to avoid any misunderstanding.

Reviewer #3

1. “The authors have not addressed my first point about what the existing literature says about the question being considered here. I'm not suggested a substantial review of the literature on early intervention but a focused consideration of what existing studies say about the question here and what is distinctive about this study that allows it to add to existing evidence.”

→Thanks for pushing on this issue. We now believe that the new literature review addresses those concerns. The paper choices were guided by reviewer #2.

2. “The authors provide some additional information on methods but they need to describe randomization - how the sequence was generated, whether there was concealment and whether there was any stratification to minimize possibility of imbalance.”

→Great point. We added a description of the randomization in Appendix C.

REVIEWER COMMENTS

Reviewer #2 (Remarks to the Author):

Thanks for addressing my comments.

Reviewer #3 (Remarks to the Author):

My comments on the last draft have been addressed and I have no further comments.

Reviewer #4 (Remarks to the Author):

Based on a series of observational studies and field experiments, this paper makes the claim that a key factor underlying socioeconomic differences in child skill development is parental beliefs, and, therefore, that changing parental beliefs is a key pathway to addressing disparities in child development. The paper's biggest contribution comes from the field experiments, which demonstrate malleability of parental beliefs and investments, and concomitant changes in child outcomes. Given that the majority of the evidence linking parental investments or interactions to child developmental outcomes is correlational, the findings from these field experiments make an important contribution.

On the other hand, the evidence that changes in parental investments and child outcomes can be traced back to changes in beliefs seems to me quite a bit weaker than the paper implies. In the newborn experiment, where the intervention focused almost exclusively on changing knowledge/beliefs, the impacts on parental investments were short-lasting and did not appear to translate to changes in child outcomes. In the home visiting experiment, where more robust impacts on parental investments and child outcomes were observed, the intervention involved much more than changing beliefs (e.g., teaching of behavioral strategies, feedback, practice, social support). While the analysis suggests that changes in parental beliefs are associated with changes in child outcomes, it is much less clear to what extent this is a critical causal factor. This is acknowledged in Appendix B but should be done more prominently in the paper given the importance for the paper's central claim.

In addition to this, I have the following comments:

1. The paper should provide a clear definition of what is meant by parental beliefs. There are many different kinds of parental beliefs and not all are considered here. This should be clarified/discussed.
2. The introduction implies that research on the origins of SES disparities in parental investments is sparse, but there is in fact a substantial literature on SES and parental beliefs in developmental psychology and in the sociological literature (e.g., see Kohn, Miller, Benasich, Lareau, Rowe; refs given at the end). This work should be acknowledged and integrated, particularly Rowe's since it directly tests

the link between parental beliefs, investments, and child outcomes in the language domain.

3. Does the analysis in Table 2 control for socioeconomic status and/or other parent characteristics?

4. I found the results section hard to read and evaluate, in large part because of the heavy reliance on footnotes and appendices to provide relevant information. I appreciate the additional analyses that are presented in the supplementary materials, but some of these analyses are in fact key to the paper's central claims (e.g., Appendix B, Appendix G) and should be summarized at least briefly in the main text.

5. It should be made clearer earlier in the paper that the control group data presented in Table 1 in fact collapses over 2 different control groups. A summary of the comparison between the intervention group and safety video control group should be given in the main text.

6. Some additional methodological details for the field experiments should be provided. How was sample size determined? What proportion of families agreed to participate?

Refs:

Rowe, M. L. (2018). Understanding Socioeconomic Differences in Parents' Speech to Children. *Child Development Perspectives*, 12(2), 122–127. <https://doi.org/10.1111/cdep.12271>

Rowe, M. L. (2008). Child-directed speech: Relation to socioeconomic status, knowledge of child development and child vocabulary skill. *Journal of Child Language*, 35, 185–205. <https://doi.org/10.1017/S0305000907008343>

Benasich, A. A., & Brooks-Gunn, J. (1996). Maternal attitudes and knowledge of child-rearing: Associations with family and child outcomes. *Child Development*, 67(3), 1186–1205.

Miller, S. A. (1988). Parents' Beliefs about Children's Cognitive Development. *Child Development*, 59(2), 259–285.

REVIEWER COMMENTS

Reviewer #2 (Remarks to the Author):

Thanks for addressing my comments.

Reviewer #3 (Remarks to the Author):

My comments on the last draft have been addressed and I have no further comments.

Reviewer #4 (Remarks to the Author):

Based on a series of observational studies and field experiments, this paper makes the claim that a key factor underlying socioeconomic differences in child skill development is parental beliefs, and, therefore, that changing parental beliefs is a key pathway to addressing disparities in child development. The paper's biggest contribution comes from the field experiments, which demonstrate malleability of parental beliefs and investments, and concomitant changes in child outcomes. Given that the majority of the evidence linking parental investments or interactions to child developmental outcomes is correlational, the findings from these field experiments make an important contribution.

On the other hand, the evidence that changes in parental investments and child outcomes can be traced back to changes in beliefs seems to me quite a bit weaker than the paper implies. In the newborn experiment, where the intervention focused almost exclusively on changing knowledge/beliefs, the impacts on parental investments were short-lasting and did not appear to translate to changes in child outcomes. In the home visiting experiment, where more robust impacts on parental investments and child outcomes were observed, the intervention involved much more than changing beliefs (e.g., teaching of behavioral strategies, feedback, practice, social support). While the analysis suggests that changes in parental beliefs are associated with changes in child outcomes, it is much less clear to what extent this is a critical causal factor. This is acknowledged in Appendix B but should be done more prominently in the paper given the importance for the paper's central claim.

→ Thank you for pointing that out. We agree and have changed the title of the manuscript, the abstract, and reformulated some of our interpretations in the main text and conclusion to acknowledge those limitations.

In addition to this, I have the following comments:

1. The paper should provide a clear definition of what is meant by parental beliefs. There are many different kinds of parental beliefs and not all are considered here. This should be clarified/discussed.

→ We clarified in the Introduction that the specific parental beliefs we study here are beliefs about how parental investments affect child skill formation and provided a more extensive definition in the introductory paragraphs of Section 2. In Sections 2.1. and 2.2., we also refer to Appendix D, where we provide a detailed description of the tool used to measure parental beliefs (SPEAK survey) with a reference to the paper presenting its development and the list of items (Suskind et al. 2018). In Appendix D.1., we mention an alternative tool (presented in Cunha et al. 2013) to measure parental beliefs about how parental investments affect child development and discusses the difference with

the beliefs measured in this paper. Appendix D.1. also presents two other measures of parental beliefs capturing self-efficacy (TOPSE survey) and perceived intelligence malleability (TOI survey) that are evaluated in Table A.4. and Table A.5.

2. The introduction implies that research on the origins of SES disparities in parental investments is sparse, but there is in fact a substantial literature on SES and parental beliefs in developmental psychology and in the sociological literature (e.g., see Kohn, Miller, Benasich, Lareau, Rowe; refs given at the end). This work should be acknowledged and integrated, particularly Rowe's since it directly tests the link between parental beliefs, investments, and child outcomes in the language domain.

→ We specified in the second paragraph of the introduction that the sparsity concerns research on the *policies* needed to mitigate the factors underpinning those SES disparities, as we believe the main originality of our analysis is to test different *interventions* aimed at shifting the beliefs of low-SES parents. We added the four below-mentioned references to our literature review, together with the recent literature on parental beliefs in economics.

3. Does the analysis in Table 2 control for socioeconomic status and/or other parent characteristics?

→ Table 2 shows linear correlations from simple linear regressions not controlling for any covariates as its main objective is to show how much of the variation in different child outcomes is captured by our belief measure alone, which is measured by the R2 of the simple linear regressions.

Nonetheless, if we re-do the computations of Table 2 controlling for parents' level of education, the correlations between beliefs and child outcomes remain large and strongly significant:

Table 3: Predictive power of parents' beliefs for children's skills

	Beliefs on impact of parents' inputs at:			
	13-16m	19-22m	25-28m	37-40m
Social-emotional skills at:				
19-22m	0.23* (0.12) 5.33%	0.16 (0.11) 3.90%		
37-40m	0.25 (0.21) 2.42%	0.39** (0.19) 5.79%	0.39* (0.20) 5.04%	0.51*** (0.18) 10.09%
Preschool language skills at:				
13-16m	0.16 (0.11) 5.61%			
19-22m	0.74** (0.32) 13.63%	0.92*** (0.31) 15.63%		
25-28m	0.95** (0.39) 13.63%	1.25*** (0.38) 15.69%	1.00** (0.43) 10.09%	
31-34m	0.75** (0.36) 13.62%	1.29*** (0.40) 18.66%	1.15** (0.44) 14.85%	
37-40m	0.79* (0.44) 19.61%	1.09*** (0.39) 21.83%	1.03** (0.44) 19.55%	0.89** (0.40) 18.99%
Peabody Picture Vocabulary at:				
37-40m	0.24** (0.12) 16.32%	0.30*** (0.11) 17.04%	0.28** (0.12) 14.46%	0.24** (0.11) 14.07%

It should be noted that we only enrolled low-SES parents in the study and therefore have little variation in terms of socioeconomic status in the sample.

4. I found the results section hard to read and evaluate, in large part because of the heavy reliance on footnotes and appendices to provide relevant information. I appreciate the additional analyses that are presented in the supplementary materials, but some of these analyses are in fact key to the paper's central claims (e.g., Appendix B, Appendix G) and should be summarized at least briefly in the main text.

→ We integrated important footnotes into the main text and added to the results section brief summaries of the analyses presented in Appendices G and H (where we moved the extensive footnote discussing experimenter demand effects), and provided additional explanations in the summary of Appendix B.

5. It should be made clearer earlier in the paper that the control group data presented in Table 1 in fact collapses over 2 different control groups. A summary of the comparison between the intervention group and safety video control group should be given in the main text.

→ We clarified that we collapse the two subgroups for the main analysis in section 2.1. when we present the experiment and added a summary of the comparison between the intervention group and safety video in section 2.2., after Table 1.

6. Some additional methodological details for the field experiments should be provided. How was sample size determined? What proportion of families agreed to participate?

→ For the newborn study, we targeted a baseline sample size of a minimum of 400 participants based on power calculations setting a target power threshold of 80%, a significance level of 5%, and assuming a 25% dropout rate. Data from a previous study by Suskind et al. (2018) were used to assess the size of expected treatment effects on parental beliefs (SPEAK survey).

For the home-visiting study, we targeted a baseline sample size of a minimum of 90 participants based on power calculations setting a target power threshold of 80%, a significance level of 5%, and assuming a 25% dropout rate. Data from our Longitudinal home-visiting study (see Leung et al. (2020)) were used to assess the size of expected treatment effects on parent-child interactions.

We included those explanations in Appendix C.1.

Regarding the second question, among the 1151 parents approached for the Newborn study, 202 refused to participate, and among the 327 parents approached for the Home-visiting study, 168 refused to participate (149 after the eligibility assessment and 19 after the preliminary assessment). Those numbers are reported in Appendix C.2., Figure A.2. Note that the proportion of participants ultimately enrolled combines families who both agreed to participate and were eligible according to our inclusion criteria (presented in Appendix C.1).

Refs:

Suskind, D. L., C. Y. Leung, R. J. Webber, A. C. Hundertmark, K. R. Leffel, I. E. Fuenmayor Rivas, and W. A. Grobman (2018). Educating parents about infant language development: A randomized controlled trial. *Clinical pediatrics* 57(8), 945–953

Leung, C. Y., M. W. Hernandez, and D. L. Suskind (2020). Enriching home language environment among families from low-ses backgrounds: A randomized controlled trial of a home visiting curriculum. *Early Childhood Research Quarterly* 50, 24–35

Refs:

Rowe, M. L. (2018). Understanding Socioeconomic Differences in Parents' Speech to Children. *Child Development Perspectives*, 12(2), 122–127. <https://doi.org/10.1111/cdep.12271>

Rowe, M. L. (2008). Child-directed speech: Relation to socioeconomic status, knowledge of child development and child vocabulary skill. *Journal of Child Language*, 35, 185–205. <https://doi.org/10.1017/S0305000907008343>

Benasich, A. A., & Brooks-Gunn, J. (1996). Maternal attitudes and knowledge of child-rearing: Associations with family and child outcomes. *Child Development*, 67(3), 1186–1205.

Miller, S. A. (1988). Parents' Beliefs about Children's Cognitive Development. *Child Development*, 59(2), 259–285.

REVIEWERS' COMMENTS

Reviewer #4 (Remarks to the Author):

My comments on the last draft have been adequately addressed. My only remaining comment is that the title now makes reference to children's "educational outcomes", but this is a stretch. This could be rephrased to "child skills", "child developmental outcomes", "child school readiness outcomes" or something similar.

Reviewer #4 (Remarks to the Author):

My comments on the last draft have been adequately addressed. My only remaining comment is that the title now makes reference to children's "educational outcomes", but this is a stretch. This could be rephrased to "child skills", "child developmental outcomes", "child school readiness outcomes" or something similar.

→ we replaced "educational outcomes" by "school readiness outcomes"